# Fast or Better? Balancing Accuracy and Cost in Retrieval-Augmented Generation with Flexible User Control

## Abstract

Retrieval-Augmented Generation (RAG) has emerged as a powerful approach to mitigate large language model (LLM) hallucinations by incorporating external knowledge retrieval. However, existing RAG frameworks often apply retrieval indiscriminately, leading to inefficiencies—over-retrieving when unnecessary or failing to retrieve iteratively when required for complex reasoning. Although recent retrieval strategies can adaptively navigate among alternative retrieval strategies, they make their selection based solely on query complexity and incorporate no mechanism for prioritizing speed over accuracy or vice versa. This lack of user-defined control makes their use infeasible for diverse user application needs. In this paper, we introduce a novel user-controllable RAG framework that enables dynamic adjustment of the accuracy-cost trade-off. Our approach leverages two classifiers: one trained to prioritize accuracy and another to prioritize retrieval efficiency. Via an interpretable control parameter $\alpha$, users can seamlessly navigate between minimal-cost retrieval and high-accuracy retrieval depending on their specific requirements. We empirically demonstrate that our approach effectively balances accuracy, retrieval cost, and user controllability [1], making it a practical and adaptable solution for real-world applications.

## 1 Introduction

Retrieval-Augmented Generation (RAG) (Lewis et al., 2020; Khandelwal et al., 2019; Izacard et al., 2023) has emerged as a promising approach to address large language models' (LLM) hallucinations, outputs that appear accurate but are actually factually incorrect. Hallucinations occur more often when the facts necessary to answer a query are outside the scope of the model's training either because the facts are too recent, too obscure or answered by proprietary data (Trivedi et al., 2022a; Yao et al., 2022). By integrating retrieval modules, RAG enables LLMs to access and incorporate relevant external information, helping them stay updated with evolving world knowledge and reducing hallucination.

Early work on retrieval-augmented LLMs focused primarily on single-hop queries (Lazaridou et al., 2022; Ram et al., 2023), where a single retrieval step retrieves relevant information based solely on the query. However, many complex queries require multi-step reasoning to arrive at the correct answer. For instance, a question such as, "When did the people who captured Malakoff come to the region where Philipsburg is located?", cannot be answered with a single retrieval step. Instead, such queries require an iterative retrieval process, in which the model retrieves partial knowledge, reasons over it, and conducts additional retrievals based on intermediate conclusions (Trivedi et al., 2022a; Press et al., 2022; Jeong et al., 2024). While multi-step retrieval enables deeper reasoning, it also incurs significant computational overhead. Conversely, for simple queries, LLMs may already encode sufficient knowledge to generate an accurate response without any retrieval whatsoever. However, many existing RAG frameworks apply retrieval indiscriminately, either over-relying on external knowledge when parametric memory suffices or failing to retrieve iteratively when deeper reasoning is required. This lack of adaptability results in inefficiencies: unnecessary retrieval increases

---

[1]Code is available at anonymous github `https://anonymous.4open.science/r/Flare-RAG-Anonymous-D6A2/`.

latency and computational cost, while insufficient retrieval leads to incomplete or incorrect answers. Given these challenges, there is a growing need for adaptive RAG systems that dynamically adjust retrieval strategies based on query complexity (Mallen et al., 2023).

The Adaptive-RAG (Jeong et al., 2024) approach attempted to balance accuracy and retrieval cost by inferring question complexity via a classifier that categorizes a query according to its most suitable retrieval strategy (e.g., no retrieval, single-step retrieval, or multi-step retrieval). But the method remains impractical due to its inability to dynamically adjust the accuracy-cost trade-off. Specifically, it lacks user-driven flexibility, preventing fine-grained control over retrieval strategies in order to support diverse application needs. In real-world scenarios, retrieval preferences vary depending on the task domain, computational constraints, and user priorities: some applications may prioritize (low) cost, while others require higher accuracy despite increased retrieval overhead.

To overcome these limitations, we propose **Flare-Aug** (**FL**exible, **A**daptive **RE**trieval-**Aug**mented Generation), a user-controllable RAG framework that enables fine-grained adjustment of the cost-accuracy trade-off. Flare-Agu introduces two classifiers: one trained to prioritize accuracy and another trained to prioritize cost. To seamlessly balance these two objectives, Flare-Aug incorporates an interpretable and easy-to-tune parameter $\alpha$, which controls the weighting between the two classifiers. This simple yet effective approach allows users to control their own accuracy and cost trade-off to best align with their specific requirements and user applications.

## 2 Related Work

Recent research on adaptive retrieval has explored various strategies to determine when and how retrieval should be performed, often leveraging internal model states or external classifiers to optimize retrieval decisions.

Several works focus on leveraging LLM internal states to adapt retrieval dynamically. Baek et al. (2024) utilize hidden state representations from intermediate layers of language models to determine whether additional retrieval is necessary for a given query. Similarly, Yao et al. (2024) extract self-aware uncertainty from LLMs' internal states to decide when retrieval should be performed. And Jiang et al. (2023) propose an iterative retrieval approach, where the model predicts the upcoming sentence and uses it as a retrieval query if the sentence contains low-confidence tokens. However, these methods require access to the LLM's internal states, making them impractical for many real-world applications, especially given the increasing reliance on proprietary, closed-source LLMs.

Another line of work trains external classifiers to predict retrieval necessity. Wang et al. (2023) train a classifier for LLM self-knowledge, allowing models to switch adaptively between retrieval and direct response generation. Jeong et al. (2024) introduce an adaptive retrieval framework that dynamically selects among no retrieval, single-step retrieval, or multi-step retrieval based on query complexity. Tang et al. (2024) propose a multi-arm bandit-based approach, where the model explores different retrieval strategies and optimizes retrieval choices based on feedback. Wang et al. (2024b) develop an adaptive retrieval framework for conversational settings, determining whether retrieval is necessary based on dialogue context. Mallen et al. (2023) take an entity-centric approach, retrieving only when the entity popularity in a query falls below a certain threshold.

Despite these advancements, none of these approaches enables direct user control over retrieval strategies.

## 3 Problem Setting: Diverse User Query Complexities and Retrieval Strategies

We begin by introducing the varying complexity of user queries and their corresponding retrieval strategies. As illustrated in Figure 1, different levels of query complexity necessitate distinct retrieval approaches. Specifically, we consider three primary retrieval strategies: no retrieval, single-step retrieval, and multi-step retrieval, which correspond to zero, one, and multiple retrieval steps, respectively. These strategies align with the availability of relevant knowledge within the model's parameters and the extent to which retrieval and reasoning must interact to generate an answer.

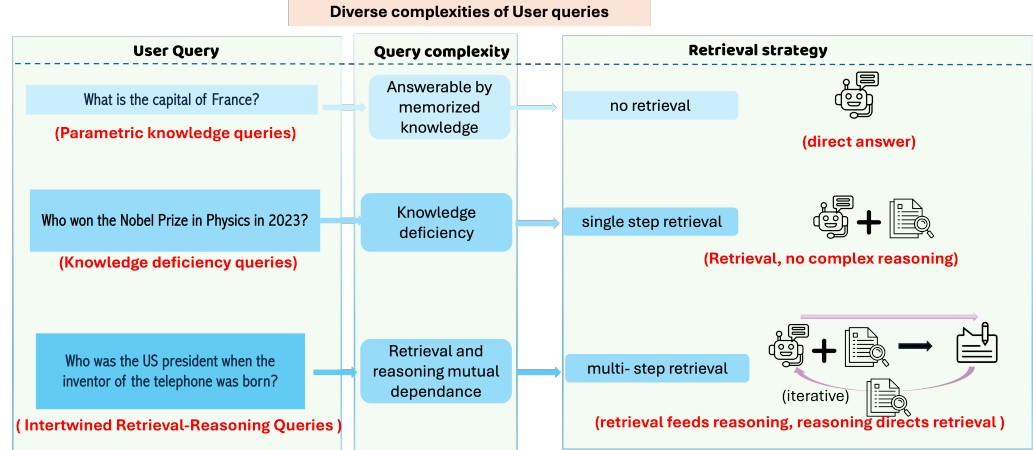

Figure 1: Illustrations of diverse user queries, their corresponding query complexity and most suitable retrieval strategy.

**Parametric Knowledge Queries & No Retrieval** Some user queries are considered simple by LLMs because they can be directly answered using their parametric knowledge—information stored within the model's parameters (Izacard et al., 2023). Since LLMs memorize widely known factual knowledge during pretraining, they can respond to such queries without relying on external retrieval. Scaling further enhances this memorization capability, improving the accuracy of responses to popular factual questions (Mallen et al., 2023). For example, queries such as *"Who wrote Romeo and Juliet?"* or *"What is the capital of France?"* can be answered accurately using only the model's internal representations. For these retrieval-free queries, retrieval augmentation is unnecessary because the model already encodes the relevant knowledge. Introducing retrieval in such cases may be redundant or even counterproductive—it can increase latency, add computational cost, or introduce noise from irrelevant retrieved documents (Su et al., 2024). Consequently, for these parametric knowledge queries that are answerable by memorized knowledge, such as well-known facts or frequently encountered knowledge, it is more efficient to use a no-retrieval strategy, allowing the model to generate responses and answer directly.

**Knowledge-Deficient Queries & Single-Step Retrieval** Some queries fall outside an LLM's parametric knowledge, such as less popular, long-tail factual knowledge that was not sufficiently represented in pretraining data (Mallen et al., 2023); time-sensitive information such as recent events and evolving regulations, or highly specialized domain knowledge that is uncommon in general corpora. For instance, a query like *"Who won the Nobel Prize in Physics in 2023?"* necessitates retrieval for models with a training cut-off date of 2022, as the event occurred afterward and is unlikely to be encoded in the model's parameters. Similarly, domain-specific questions such as *"What are the latest FDA guidelines for AI-based medical devices?"* may not be well covered in publicly available pretraining datasets, requiring retrieval from authoritative sources to provide an accurate and up-to-date response. We refer to these queries as Knowledge Deficient Queries, where retrieval is essential to supplement missing knowledge, but complex reasoning is not required. In these cases, retrieval serves as a direct knowledge lookup, providing factual information that the model can integrate into its response without needing complex reasoning or inference. Once the relevant information is retrieved, the model can directly generate an accurate answer.

**Intertwined Retrieval-Reasoning Queries & Multi-Step Retrieval** Some complex queries require both retrieval and reasoning in an interdependent manner, where the model must retrieve information iteratively while reasoning at each step to determine the next retrieval. Unlike Knowledge Deficient queries, where passively consuming content from the single retrieval step is sufficient to reach the correct answer, Intertwined Retrieval-Reasoning Queries demand both factual knowledge and the model's ability to reason, plan, infer, and integrate retrieved facts to iteratively refine its understanding and construct the final correct answer. For example, in response to the query *"Who was the US president when the inventor of the telephone was born?"*, the model must first recognize that additional information is needed about the inventor of the telephone. This reasoning step directs

the retrieval process, leading the model to retrieve the name *Alexander Graham Bell* and his birth year. Once retrieved, this new information updates the model's understanding, allowing it to reason further and determine that it must now retrieve the name of the U.S. president during the retrieved year. This retrieval feeds reasoning, and reasoning directs retrieval, forming a continuous cycle that progresses until the final answer is constructed. A single retrieval step is likely to fail in such cases because the initially retrieved information alone is insufficient to answer the query. The sequential nature of retrieval in these queries requires the model to dynamically decides what to retrieve at each step, ensuring that each retrieval builds upon prior reasoning to progressively arrive at the correct answer. However, compared to no retrieval and single step retrieval, this multi-step retrieval process incurs higher computational cost, as each additional retrieval step increases latency and resource consumption.

## 3.1 Adaptive Retrieval

**The Need for Adaptive Retrieval in Real-World Applications** In real-world applications, user queries vary significantly. A fixed retrieval strategy can lead to unnecessary computational costs, increased latency, or suboptimal accuracy, depending on the characteristics of the query and no particular retrieval strategy is optimal for all the query types. For example, forcing retrieval for simple queries that can be answered directly from an LLM's parametric knowledge (e.g., *"Who wrote Pride and Prejudice?"*) wastes computational resources and increases latency without improving accuracy. Conversely, for time-sensitive queries (e.g. *"what is the current inflation rate in the U.S.?"*), relying solely on parametric knowledge leads to outdated or hallucinated responses, while employing multi-step retrieval unnecessarily increases cost and latency. Similarly, for complex queries that require multi-step reasoning, some level of latency is unavoidable. For these queries, using no retrieval or only a single-step retrieval would often result in incomplete or incorrect responses, reducing the utility and overall effectiveness of the system. Thus, it is crucial to develop an adaptive retrieval system that can dynamically balances accuracy, efficiency, cost.

**Limitations of Prior Adaptive Retrieval Approaches** While previous work, such as Adaptive-RAG (Jeong et al., 2024), has attempted to balance efficiency and cost by training classifiers using data collected from model predictions and the inductive biases in the datasets themselves, they are adaptive only to static query complexities but not to user preferences, making them insufficient for handling the diverse needs of real-world applications. A truly adaptive retrieval system should incorporate both query complexity and user-level controllability. In addition to diverse query complexities, different users and application settings may prioritize accuracy over cost or vice versa, depending on their specific requirements. For example, a medical researcher using an LLM to summarize clinical trial results may prefer higher accuracy and completeness, even at the cost of increased retrieval latency and computational expenses. In contrast, a customer service chatbot handling frequent, simple user queries (e.g., *"What is the return policy?"*) might prioritize fast, cost-effective responses, as occasional inaccuracies pose only negligible risk. Similarly, a stock market analyst requiring precise, up-to-date financial information might opt for a retrieval-intensive approach, despite longer response times, whereas a real-time virtual assistant answering everyday factual queries should minimize retrieval overhead, emphasizing speed over exhaustive accuracy. Given these diverse application domains and varying user demands, a one-size-fits-all adaptive retrieval approach is still suboptimal. Instead, a truly adaptive RAG system should be both efficient and customizable, enabling dynamically optimized trade-offs between accuracy and cost while allowing flexible user control.

## 4 Flare-Aug: Flexible Adaptive Retrieval Augmented Generation

In this section, we present the main components of our framework, a Cost-Optimized Classifier, which is dynamic and LLM-dependent, and a Reliability-Optimized Classifier, which is static and dataset-dependent. An overview of the Flare-Aug framework is illustrated in Figure 2 and described in the sections below.

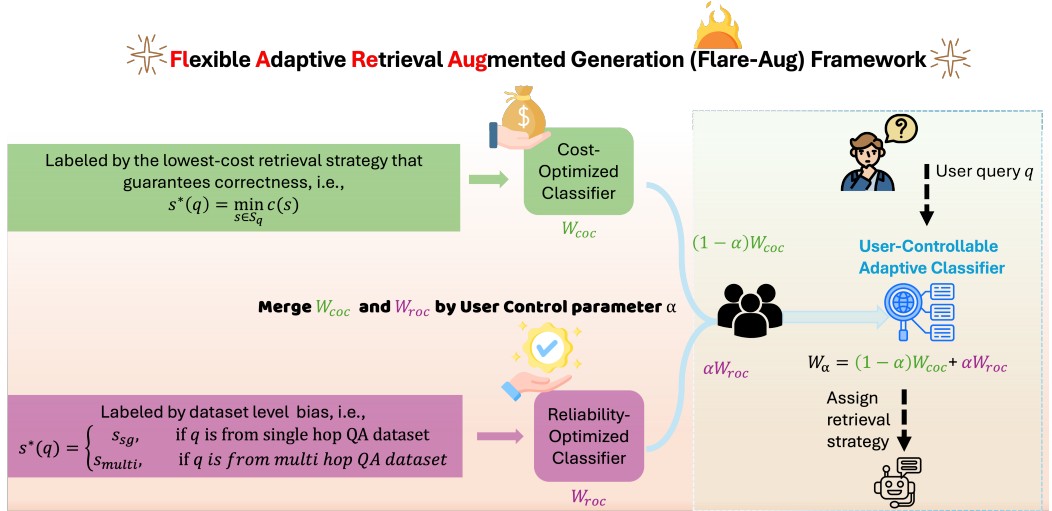

Figure 2: An overview of Flexible Adaptive Retrieval Augmented Generation (Flare-Aug) framework.

## 4.1 COST-OPTIMIZED CLASSIFIER

The goal of the Cost-Optimized Classifier is to select for a given query the cheapest retrieval strategy that still ensures a correct answer. This classifier is LLM-dependent, since the training data is collected specifically from the LLM used for retrieval, and the prediction is sensitive and tailored to the LLM. The *<query, retrieval strategy>* pairs that comprise the training data are created by using the LLM to answer with all the three strategies–no retrieval, single-step retrieval and multi-step retrieval: the gold label for each query is the minimal cost strategy that produces a correct answer. Thus, the labeling process itself is biased towards maximally eliminating unnecessary retrieval. The trained classifier, as a result, minimizes retrieval cost, latency, and computational complexity while maintaining an acceptable level of accuracy. (See the experiments in later section when $\alpha = 0$). However, a key concern for this classifier is that the LLM may arrive at the correct answer by chance or by exploiting shortcuts in reasoning, leading to unverifiable answers. While this classifier is highly beneficial for cost-sensitive and efficiency-driven applications, it is unsuitable for high-risk domains where retrieval must be systematic and reliably grounded in external knowledge.

**Training Data for Cost-Optimized Classifier** The labeling process for the cost-optimized classifier is straightforward: we aim to select the cheapest retrieval strategy that still ensures a correct answer. Formally, let $S = \{s_{no}, s_{sg}, s_{multi}\}$ denote the set of available retrieval strategies: no retrieval ($s_{no}$), single-step retrieval ($s_{sg}$) and multi-step retrieval ($s_{multi}$). For each query $q$ in the training data, we evaluate whether the LLM answers it correctly under each retrieval strategy. Denote $LLM(q, s)$ as the correctness function, where:

$$LLM(q, s) = \begin{cases} 1, & \text{if the LLM answers } q \text{ correctly with retrieval strategy } s, \\ 0, & \text{otherwise.} \end{cases}$$

Let $S_q \subseteq S$ be the subset of retrieval strategies that lead to correct answers for $q$: $S_q = \{s \in S | LLM(q, s) = 1\}$. Since the retrieval strategies incur different costs, where $c(s_{no}) < c(s_{sg}) < c(s_{multi})$, we assign the lowest-cost correct strategy as the label for $q$:

$$s^*(q) = \arg \min_{s \in S_q} c(s)$$

For example, if $S_q = \{s_{no}, s_{sg}\}$, we choose $s_{no}$ as the label for $q$ since it has a lower cost than $s_{sg}$. Thus, for each query, we first determine $S_q$, the set of all retrieval strategies that yield a correct answer. We then assign $s^*(q)$ as the strategy with the lowest retrieval cost. If $S_q = \emptyset$ (i.e., the LLM fails to answer the query correctly under any strategy), we exclude the query from the dataset.

## 4.2 RELIABILITY-OPTIMIZED CLASSIFIER

The goal for the Reliability-Optimized Classifier is to assign a retrieval strategy to a query based on intrinsic difficulty of the query itself, ensuring consistency, stability and reliability across different LLMs. This classifier is trained on <*query, retrieval strategy*> pairs with binary labels: single-step retrieval and multi-step retrieval, using data from a single-hop QA dataset and multi-hop QA dataset, respectively. Here we rely on the assumption that the single hop QA dataset is in fact intrinsically biased towards questions that require single-step retrieval, and the multi-hop QA dataset towards questions that require multi-step retrieval. Since there are only two labels, this classifier guarantees least one retrieval, and may induce unnecessary computational overhead when using more powerful LLMs or handling straightforward queries that do not require retrieval. Despite this, since it is not LLM specific, it is more stable and reliable, encouraging higher accuracy and more reliable response. (See experiments in later section when $\alpha = 1$). This classifier is effective in applications where accuracy is paramount, albeit at a higher cost.

**Training Data for Reliability-Optimized Classifier** The Reliability-Optimized Classifier is trained using binary labels—single-step retrieval and multi-step retrieval—derived from existing question-answering datasets. Unlike the Cost-Optimized Classifier, which tailors the retrieval strategy to a specific LLM, this classifier relies on dataset-level biases to infer retrieval needs, making it more stable across different LLMs. For each query $q$, we assign a label $s^*(q)$ based on the dataset it belongs to:

$$s^*(q) = \begin{cases} s_{sg}, & \text{if } q \text{ originates from a single-hop QA dataset} \\ s_{multi}, & \text{if } q \text{ originates from a multi-hop QA dataset.} \end{cases}$$

This labeling approach assumes that queries from single-hop QA datasets are best handled with single-step retrieval, while multi-hop QA datasets require multi-step retrieval. Since the classifier does not verify correctness based on a specific LLM, it ensures LLM-agnostic stability though it does not minimize unnecessary retrieval.

## 4.3 USER-CONTROLLABLE ADAPTIVE CLASSIFIER

To fully realize user-controllability, we introduce a simple and interpretable parameter $\alpha$ that allows users to define their preferred trade-offs, providing flexibility across diverse use cases while dynamically optimizing the retrieval strategy. Specifically, after training the Cost-Optimized Classifier (denoted by $W_{coc}$) and the Reliability-Optimized Classifier (denoted as $W_{roc}$), we construct a User-Controllable Adaptive Classifier by interpolating between their parameters based on the user-defined control parameter $\alpha$. The resulting classifier is defined as: $W_\alpha = (1 - \alpha)W_{coc} + \alpha W_{roc}$, where $\alpha \in [0, 1]$ controls the trade-off between cost and accuracy. Setting $\alpha = 0$ results in a purely cost-optimized classifier, while $\alpha = 1$ yields a fully reliability-optimized classifier. This formulation provides a continuous and customizable retrieval strategy, allowing users to flexibly adjust the balance between efficiency and accuracy based on their specific needs.

## 5 EXPERIMENTS

### 5.1 EXPERIMENTAL SETUP

We describe the datasets, baselines, evaluation metrics, and experimental details in our study.

**Dataset** To simulate a realistic retrieval scenario, where queries naturally vary in complexity and diversity, we evaluate on a mixed dataset with queries from three single-hop question answer datasets—SQuAD (Rajpurkar, 2016), Natural Questions (Kwiatkowski et al., 2019), and TriviaQA (Joshi et al., 2017)—and three multi-hop QA datasets—MuSiQue (Trivedi et al., 2022b), HotpotQA (Yang et al., 2018), and 2WikiMultiHopQA (Ho et al., 2020). This combination reflects the heterogeneous nature of real-world queries, encompassing both fact-based retrieval tasks and multi-hop reasoning tasks that require linking information across multiple sources. For training, we randomly sample 500 queries from each dataset, resulting in a total of 3,000 training examples. Similarly, for evaluation, we construct a test set by randomly sampling 500 queries from each dataset, yielding a total of 3,000 test queries. **Baselines** To assess the effectiveness of our approach in balancing accuracy, cost, and user control flexibility, we compare it against both adaptive and static retrieval

strategies. Specifically, we evaluate our method against the adaptive retrieval approach **Adaptive-RAG** (Jeong et al., 2024), as well as three static retrieval strategies: always use **No Retrieval**; always use **Single-Step Retrieval** and always use **Multi-Step Retrieval** (Trivedi et al., 2022a).

**Evaluation Metrics** We evaluate the trade-off between accuracy and retrieval cost using two metrics. *Accuracy* measures whether the predicted answer contains the ground-truth answer. *Retrieval cost* is quantified as the number of retrieval steps required to generate an answer, where No Retrieval has a cost of 0, Single-Step Retrieval has a cost of 1, and Multi-Step Retrieval incurs a variable cost depending on the number of retrieval steps per query. By jointly considering these two metrics, we assess how different retrieval strategies balance the trade-off between response quality and computational efficiency. Note that we use retrieval-side cost as a proxy for cost due to its Interpretability and simplicity: Retrieval step count is a transparent and easily measurable proxy that correlates with latency, token consumption, and compute in RAG systems. Unlike in math reasoning tasks—where reasoning token length is often used as a proxy for cost—the context length in RAG is not a reliable cost metric. This is because context size in RAG is largely dictated by the retrieval configuration (e.g., top-k) rather than the intrinsic complexity of the query. For example, retrieving 10 passages versus 2 directly affects context length without reflecting deeper reasoning complexity. In Appendix A.1, we further justified the choice of these two evaluation metrics.

**Implementation Details** Following Jeong et al. (2024) and Trivedi et al. (2022a), we use BM25, a term-based sparse retrieval model, as the retriever[2]. For queries from multi-step datasets, we utilize the pre-processed corpus from Trivedi et al. (2022a) as the external document corpus, while for queries from single-hop datasets, we use Wikipedia, preprocessed by Karpukhin et al. (2020), as the external document corpus. For answer generation [3], we employ two sizes of Flan-T5 models (Chung et al., 2024) (Flan-T5 XL and Flan-T5 XXL) and two sizes of GPT-4o models (Hurst et al., 2024) (GPT-4o and GPT-4o-mini). For the classifier, we train a T5-Large model (Raffel et al., 2020) using 500 samples from each dataset, resulting in a total of 3,000 queries for the labeling process. The classifier is trained with a learning rate of 3e-5 and a batch size of 64 on a single 80GB A100 GPU. We train the model for 20 epochs, as the loss has converged stably without indications of overfitting [4]. We demonstrate the robustness of the classifier in Appendix B, with experiments on different training epochs and different sizes of classifiers.

## 5.2 MAIN RESULTS

Figure 3 and Figure 4 illustrate the accuracy and retrieval cost across different values of $\alpha$. We observe that both accuracy and cost **monotonically** increase with the user-controllable parameter $\alpha$, allowing users to easily adjust the retrieval strategy based on their preferred trade-offs between accuracy and efficiency. Due to space limitations, the exact quantitative results can be found in Table 3 in Appendix C.

**Accuracy Trends Across Different** $\alpha$ As shown in Figure 3, the No Retrieval strategy yields the lowest accuracy, with values below 0.2 for the Flan-T5 models, while achieving relatively higher performance GPT4o series models (approximately 0.43 for GPT-4o Mini and 0.53 for GPT-4o). Applying Single-Step Retrieval significantly improves accuracy, particularly for Flan-T5 models, increasing their accuracy to around 0.39. However, the performance gains for GPT-4o models are relatively smaller, as these models already exhibit strong parametric knowledge storage and advanced reasoning abilities. In contrast, Flan-T5 models benefit more from retrieval, as they are less capable of answering queries without external information. The accuracy of Adaptive-RAG falls between Single-Step Retrieval and Multi-Step Retrieval. Meanwhile, our user-controllable

---

[2]While BM25 is a sparse retriever, it remains a strong and widely-used baseline in the adaptive RAG approaches such as Jeong et al. (2024); Tang et al. (2024); Jiang et al. (2023); Su et al. (b); Wang et al. (2024a; 2023); Trivedi et al. (2022a); Mallen et al. (2023). Studies (e.g., Ram et al. (2023)) show that BM25 continues to outperform many dense retrievers.

[3]We used the same prompting format as in prior work Trivedi et al. (2022a); Jeong et al. (2024), the exact prompts we used are publicly available in the codebase associated with Trivedi et al. (2022a), which can be accessed at: https://github.com/StonyBrookNLP/ircot/tree/main

[4]Both classifiers are lightweight and fast to train. For instance, training the Cost-Optimized Classifier on GPT-4o-mini–specific data (1,377 samples, 450 steps) takes only 223 seconds with a batch size of 64. The Reliability-Optimized Classifier (trained on 2,400 samples, 750 steps) takes 417 seconds. In total, training both classifiers requires approximately 640 seconds. We refer readers to discussions in Appendix A.2.

approach achieves accuracy levels ranging between those of Single-Step Retrieval and Multi-Step Retrieval, allowing users to tailor the retrieval strategy to different application needs. Notably, our approach can even outperform Multi-Step Retrieval for Flan-T5-XL, Flan-T5-XXL, and GPT-4o Mini, demonstrating its effectiveness in optimizing retrieval decisions based on query complexity.

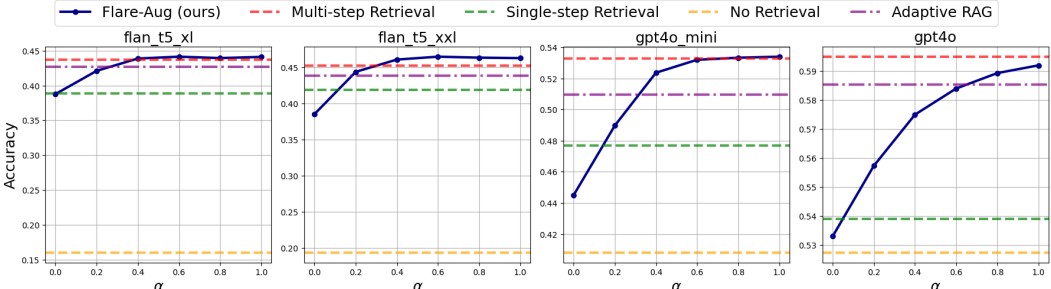

Figure 3: Average accuracy of different approaches on validation set.

**Cost Trends Across Different** $\alpha$ Figure 4 illustrates the retrieval cost, measured in retrieval steps, across different values of the user-controllable parameter $\alpha$. Similar to the accuracy trends observed in Figure 3, retrieval cost monotonically increases with $\alpha$, enabling users to adjust retrieval expenditure according to their cost constraints. For Flan-T5 models, the minimum retrieval cost (i.e., setting $\alpha = 0$) is close to Single-Step Retrieval, indicating that these models require certain amount of retrieval to answer queries correctly. In contrast, for GPT-4o models, the minimum cost is closer to No Retrieval, as these models possess stronger parametric knowledge and reasoning abilities, allowing them to correctly answer a larger proportion of queries without retrieval. This behavior highlights the adaptability of the Cost-Optimized Classifier, which is LLM-specific and automatically adjusts to the retrieval needs of different models—retrieving more for Flan-T5 models while retrieving less for GPT-4o models. Furthermore, while Figure 3 shows that our approach achieves accuracy comparable to or exceeding Multi-Step Retrieval as $\alpha$ increases, Figure 4 demonstrates that its retrieval cost remains consistently lower than that of Multi-Step Retrieval. Thus, compared to Multi-Step Retrieval, our approach incurs significantly lower retrieval overhead while without sacrificing accuracy. **Accuracy-Cost trade-off** In Figure 5, accuracy-cost trade-off of our approach

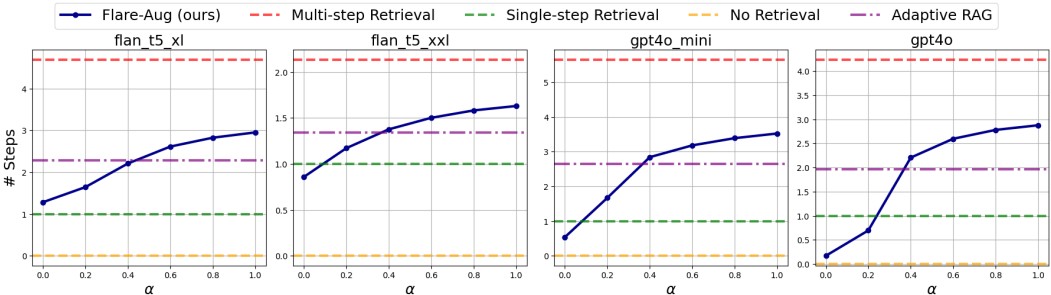

Figure 4: Total number of steps (cost) of different approaches on validation set.

across different values of $\alpha$. For Flan-T5 XL, Flan-T5 XXL, and GPT-4o Mini, Adaptive-RAG is positioned on the lower right side of our trade-off curves, indicating that our approach consistently outperforms Adaptive-RAG in both accuracy and cost efficiency. Another way to interpret the accuracy-cost plot is by fixing either accuracy or cost and comparing the other metric. For example, when fixing accuracy to the level achieved by Multi-Step Retrieval, our approach requires significantly fewer retrieval steps, demonstrating its ability to achieve comparable or superior accuracy with lower retrieval overhead.

## 5.3 PRACTICALITY OF SETTING $\alpha$

To further demonstrate the practicality of our approach, we provide two simple and intuitive strategies that can be used for setting $\alpha$:

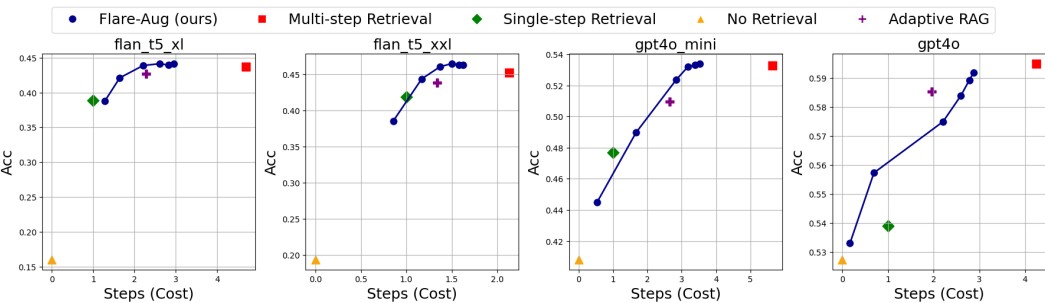

Figure 5: Accuracy-Cost plot of different approaches on validation set.

- **Incremental Adjustment**: Since $\alpha$ increases monotonically with both cost and accuracy, users can start with an initial $\alpha$ and refine it based on observed retrieval cost and response quality. If the retrieval cost exceeds their budget, they can lower $\alpha$; if the response quality is unsatisfactory, they can increase $\alpha$.
- **Validation-Based Estimation**: Users can estimate a suitable $\alpha$ directly by examining the accuracy-cost trade-off on the validation set, allowing them to select a value that aligns with their desired balance between retrieval cost and accuracy. In Appendix C.2, we conduct additional experiments on the test set to illustrate that $\alpha$ can be easily estimated based on results on validation set.

## 6 CONCLUSION

In this work, we present Flare-Aug, a flexible and adaptive retrieval-augmented generation framework that incorporate user controllability into RAG framework besides accuracy and cost trade-offs. Unlike existing adaptive RAG methods that provide only static adaptation to query complexity, Flare-Aug enables fine-grained and dynamic control over retrieval behavior, allowing users to balance accuracy and cost based on their specific needs. By enabling truly adaptive retrieval tailored to user preferences, Flare-Aug represents a step toward more efficient, customizable, and scalable retrieval-augmented generation, paving the way for future advancements in personalized RAG system.

## 7 LIMITATIONS AND FUTURE WORK

Future research could extend our approach to more advanced adaptive retrieval-augmented generation systems that dynamically adjust not only retrieval strategies but also LLM selection based on task complexity and resource constraints. Additionally, while our classifier is computationally lightweight compared to retrieval costs, an important direction for future work is to develop a more unified retrieval adaptation framework, enabling LLMs to self-regulate retrieval decisions without relying on an external classifier. Though in our work, we only have three class. In real-world deployments, where queries may be unanswerable, it would be preferable to include an additional unanswerable class.

Adapting Flare-Aug to multi-turn or dynamic user contexts is an important and promising direction for future work. While our current formulation assumes single-turn queries, we envision several ways in which Flare-Aug could be extended to multi-turn dialogue systems: (1) Context-aware classification: The retrieval strategy classifier could be conditioned not only on the current query, but also on dialogue history (e.g., previous user intents, prior retrieved content), enabling it to better estimate query complexity in conversational flow. (2) Dynamic $\alpha$ adaptation: Instead of a fixed $\alpha$, the system could adjust $\alpha$ over the course of a dialogue, depending on user behavior (e.g., impatience vs. information-seeking), query difficulty, or accumulated system confidence. (3) User profile integration: In persistent user-facing systems, $\alpha$ could be personalized based on explicit preferences (e.g., set by the user) or learned from historical interaction patterns. These extensions would allow Flare-Aug to support more nuanced retrieval strategies in conversational assistants, tutoring systems, or customer support scenarios, where both query complexity and user tolerance for cost may evolve over time.

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

## A    ADDITIONAL DISCUSSIONS

### A.1    JUSTIFICATION OF EVALUATION METRICS

In this work, we mainly focus on retrieval-side cost and abstract away generation-related costs (such as the size of the LLM) by fixing the LLM backbone. This design choice allows us to isolate the retrieval component for targeted optimization, rather than overlapping with existing work on LLM routing and resource-aware decoding (e.g., Ong et al. (2024); Wang et al. (2025); Chuang et al.; Stripelis et al. (2024)), which primarily focus on switching between LLMs or managing generation behaviors. Our approach is complementary to those methods and can be integrated into broader adaptive systems that jointly optimize both retrieval and generation paths. We model retrieval cost via the number of retrieval steps, which directly impacts latency, bandwidth, and context window usage, while remaining easy to quantify and interpret. This abstraction enables a clear and actionable trade-off between accuracy and efficiency—central to our user-controllable framework. Different from math reasoning, where the cost is easily associated with the context length, in the RAG setting, the context length depends on the passages retrieved (the parameter for RAG), which might not reflect the actual cost. While our current focus is on retrieval cost, Flare-Aug is compatible with and can be extended to systems that also adapt LLM usage.

### A.2    COMPUTATIONAL OVERHEAD OF THE CLASSIFIER

Our approach introduces minimal computational overhead. Both classifiers are lightweight and fast to train. For example, training the Cost-Optimized Classifier on model-specific data (3,000 samples, 20 epochs, batch size 64) takes approximately 478.75 seconds on average across four different LLMs. The Reliability-Optimized Classifier, which is LLM-agnostic and trained once, requires only 417 seconds. In total, training both classifiers takes an average of 895.75 seconds (Table 1). Despite using an additional classifier compared to Adaptive-RAG, our method does not incur extra training cost, as Adaptive-RAG also trains a classifier, combining both dataset-level and LLM-specific signals. Thus, the total volume of training data is the same, and the end-to-end training time is comparable. Moreover, Adaptive-RAG must retrain its classifier whenever the LLM changes, while our Reliability-Optimized Classifier can be reused across models. This makes our approach more efficient in practice, especially in multi-model deployment settings. Furthermore, since all classifiers operate on short query inputs only, the overall training and inference cost remains low. The training time can be further reduced by increasing the batch size or subsampling the training data, making our method suitable for real-world, resource-constrained scenarios.

| Reliability-Optimized Classifier | Cost-Optimized Classifier | | | | Average Total |
|---|---|---|---|---|---|
| | flan-t5-xl | flan-t5-xxl | gpt4o | gpt4o-mini | |
| 417 | 479 | 479 | 479 | 478 | 895.75 |

Table 1: Classifier training time using batch size (in seconds) using 3000 training data with a batch size 64 for 20 epochs on t5-large model. The average total is the total training time for 2 classifiers, where the time for cost-optimized classifier is averaged over LLM-specific data. The training time can be further reduced by increasing the batch size or reducing the number of training data.

### A.3    IS HAVING THE PARAMETER $\alpha$ A DRAWBACK?

Tuning the parameter $\alpha$ is straightforward and does not require "careful calibration." The parameter induces a monotonic trade-off between retrieval cost and accuracy (as shown in Figures 3, 4, 5), making it easy to tune with minimal validation. Moreover, unlike Adaptive-RAG, Flare-Aug empowers users with explicit control over the cost-accuracy trade-off—a valuable feature in real-world applications with varying preferences and constraints. Thus, this controllability via $\alpha$ is an advantage, not a drawback.

### A.4    POTENTIAL NOISE AND BIAS INDUCED BY DATASET LEVEL LABELS

The dataset-level labels might introduce some degree of noise, as not all single-hop or multi-hop queries strictly conform to single-step or multi-step retrieval needs. However, we find that this la-

beling approach is effective in practice. As shown in Figures 3, 4, 5, our method achieves strong performance despite this potential noise, indicating that the Reliability-Optimized Classifier is robust to label imperfections. Moreover, we intentionally leverage the inductive bias of existing datasets: single-hop datasets are typically biased toward queries solvable with a single retrieval step, while multi-hop datasets are biased toward multi-step reasoning. This bias serves as a useful supervisory signal, enabling us to train a reliable classifier without requiring costly manual annotation. While we acknowledge that a more fine-grained labeling scheme may further improve precision, it would require substantial relabeling efforts and undermine the lightweight and scalable nature of our framework. After weighing this trade-off, we believe our current strategy—using dataset-level labels—is a practical and effective choice, offering strong performance with minimal overhead.

## A.5 Main Advantage Comparing with Adaptive RAG

While our work shares the high-level motivation of balancing accuracy and cost with Adaptive-RAG, our contribution goes beyond incremental performance improvement. Note that adaptive rag is not truly adaptive since it doesn't adapt to different users without retraining the classifier. The core novelty of Flare-Aug lies in introducing explicit user controllability into the retrieval process—something Adaptive-RAG lacks.Adaptive-RAG makes a fixed trade-off between accuracy and cost on behalf of the user. However, different users and applications may have different priorities: some may tolerate higher latency for better accuracy (e.g., clinical summarization), while others may prefer faster, lower-cost answers even at the expense of slight performance degradation (e.g., customer service bots). Our method allows users to dynamically adjust the trade-off according to their specific goals or constraints. In this sense, Flare-Aug is designed for cost-aware personalization, rather than optimizing for a one-size-fits-all adaptive retrieval.

## A.6 Adding an Additional "Unanswerable" Class

For the purposes of comparison with Adaptive-RAG, we followed the same convention used in that work and trained the classifier only on answerable queries. This ensures a clean and fair comparison, as Adaptive-RAG does not handle the "unanswerable" case either. That said, our framework is easily extensible to support this scenario. Incorporating a fourth class would require only minimal changes to the data labeling and classifier training process. In Table 2, we show the result fo Flare-RAG with an additional "unanswerable" class, where we introduced an additional "unanswerable" class when training the Cost-Optimized Classifier, resulting in a 4-class classification setting. Specifically, a query is labeled as unanswerable if none of the available strategies (across cost levels) lead to a correct answer—i.e., all fail. For such cases, unanswerable queries should be handled by having the system either express uncertainty (e.g., by saying "I don't know" (Deng et al., 2024; Zhang et al., 2024)) or defer to human experts (Su et al., a). However, as our work does not operate in a setting that supports answer refusal or human-in-the-loop escalation, these options are not actionable. From a cost-optimization perspective, we treat the unanswerable class as a signal that high-cost reasoning is unlikely to help. In such cases, the system should default to the cheapest strategy, as both cheap and expensive strategies are similarly ineffective. However, in Table 2, we observed that the performance deteriorates in this setup. Specifically, the newly added "unanswerable" class constitutes a large portion of the training data, which shifts the classifier's focus. As a result, the Cost-Optimized Classifier—originally effective at distinguishing between retrieval strategies (e.g., no retrieval, single-step, and multi-step)—becomes less discriminative among them. Instead, it becomes primarily focused on learning to differentiate between answerable and unanswerable queries, which undermines its utility for fine-grained cost-aware strategy selection.

## A.7 Justification of the Importance of Having Both Classifiers

The existence of the two classifiers provides users with the ability to choose or interpolate between them based on their application needs.The preferred classifier depends on both query type and user preference: In high-stakes domains such as medical or legal contexts, users may prioritize accuracy and reliability, even at the cost of latency. In these settings, the Reliability-Optimized Classifier (which favors more cautious retrieval) is more appropriate. In low-risk or latency-sensitive applications (e.g., casual assistants, customer support), users may be willing to tolerate minor drops in accuracy for a faster and cheaper response. In these cases, the Cost-Optimized Classifier is preferable. Note that both classifiers alone are already good enough in balancing accuracy and latency,

| Models | flan-t5-xl | | flan-t5-xxl | | gpt4o-mini | | gpt4o | |
|---|---|---|---|---|---|---|---|---|
| | Acc | Steps | Acc | Steps | Acc | Steps | Acc | Steps |
| $\alpha = 0.0$ | 0.184 | 0.054 | 0.216 | 0.065 | 0.408 | 0.0 | 0.527 | 0.0 |
| $\alpha = 0.2$ | 0.345 | 0.728 | 0.37 | 0.704 | 0.427 | 0.329 | 0.531 | 0.058 |
| $\alpha = 0.4$ | 0.424 | 1.705 | 0.452 | 1.259 | 0.515 | 2.465 | 0.569 | 1.94 |
| $\alpha = 0.6$ | 0.435 | 2.273 | 0.461 | 1.429 | 0.524 | 2.982 | 0.58 | 2.46 |
| $\alpha = 0.8$ | 0.438 | 2.666 | 0.462 | 1.545 | 0.532 | 3.259 | 0.585 | 2.696 |
| $\alpha = 1.0$ | 0.439 | 2.916 | 0.461 | 1.621 | 0.533 | 3.425 | 0.587 | 2.817 |

Table 2: Result of extending Flare-RAG with an additional "unanswerable" class.

however, we still want to give users more control over where they want to land on these already good enough parameter spaces.

The existence of two classifiers also reflects a multi-objective learning perspective. Specifically, our system decomposes the overall decision problem into two distinct sub-goals: (1) minimizing cost (latency, token usage), and (2) maximizing reliability (robust correctness). These sub-goals are inherently different and are best addressed by specialized experts: The Cost-Optimized Classifier is trained in an LLM-specific manner because its role is tightly coupled with the inference behavior of a particular model. LLM-dependent training ensures that it can make fine-grained distinctions based on model characteristics and capability. The Reliability-Optimized Classifier is trained on LLM-agnostic data to ensure generalization across different model behaviors, aligning with conservative decision-making goals regardless of the deployed model.

## B  ABLATION ON CLASSIFIER

**Different sizes of classifiers**  Figure 6 shows accuracy and total steps predicted by different sizes of classifier and $\alpha$ on validation set, and Figure 7 shows a more clear comparison on the trade-offs of accuracy and cost by different sizes of classifiers. Though the performance of different classifiers varies, we find that there is no dominating advantage in using larger classifiers.

**Different training epochs**  Figure 8 shows accuracy and total steps predicted by t5-3b and t5-base training for 20 and 40 epochs on validation set, and Figure 9 shows a more clear comparison on the trade-offs of accuracy and cost by t5-3b and t5-base training for 20 and 40 epochs on validation set.

**Training Loss**  In Figure 10, we show average training loss v.s. training epochs for cost optimized classifier for different models and reliability optimized classifier.

## C  MORE EXPERIMENTAL RESULTS

### C.1  QUANTITATIVE RESULTS ON VALIDATION SET

In Table 3, we detail the step and accuracy of different retrieval strategies on validation dataset.

### C.2  FEASIBILITY OF ESTIMATING $\alpha$ BASED ON VALIDATION SET.

In Figure 11, Figure 12 and Figure 13, we show the accuracy and cost with various $\alpha$, as well as the accuracy-cost plot on test set. (The quantitative result in Table 4). Combines with the experimental results for validation dataset, we empirically verify that users can estimate the accuracy-cost trade-off from validation results. As shown in Figure 11, Figure 12 and Figure 13, the test set follows the same trend as the validation set. For example, based on Figure 3, setting $\alpha = 0.4$ or larger for flan-t5-xl achieves an accuracy level comparable to Multi-Step Retrieval, and using the same $\alpha$ value on test set, as shown in Figure 11, yields a similar accuracy-cost balance. Moreover, if the goal is to achieve an accuracy higher than Adaptive-RAG, the validation results suggest setting $\alpha = 0.2, 0.2, 0.4, 0.6$ for flan-t5-xl, flan-t5-xxl, gpt-4o-mini and gpt-4o respectively, which is consistent

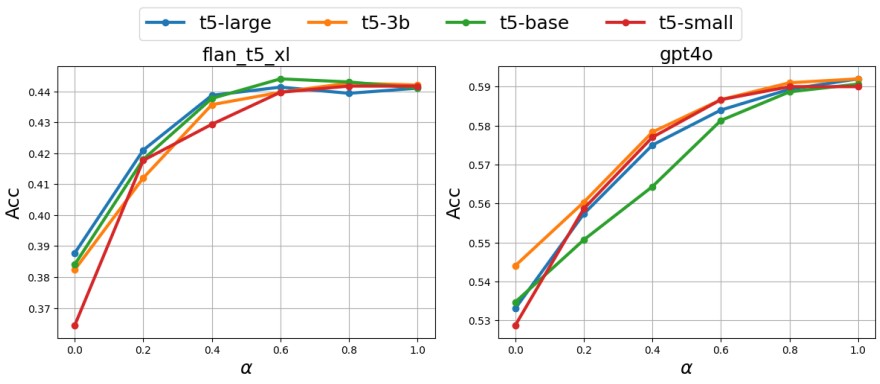

(a) Accuracy using different parameter $\alpha$ and different sizes of classifiers.

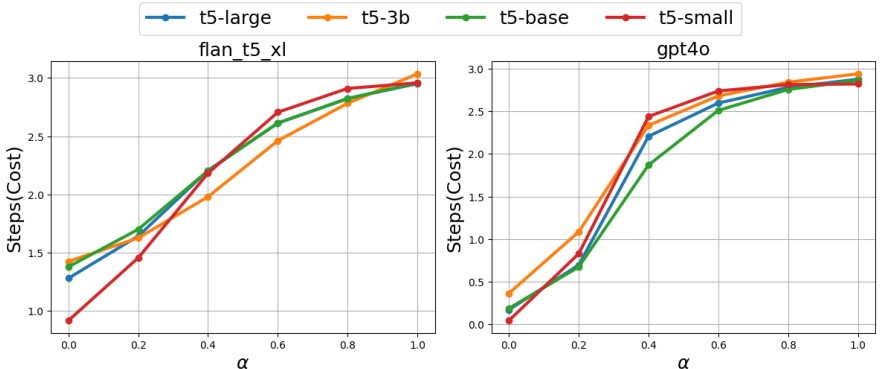

(b) Total retrieval steps (cost) using different parameter $\alpha$ and different sizes of classifiers.

Figure 6: Accuracy and total steps predicted by different sizes of classifier and $\alpha$ on validation set.

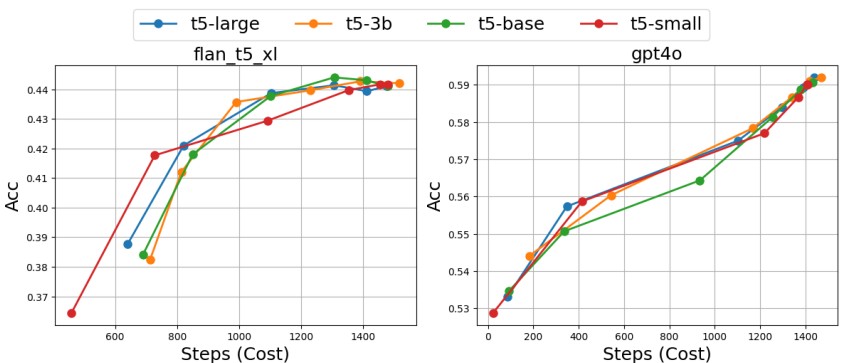

Figure 7: Accuracy-Cost trade-off plot of different sizes of classifiers.

with that on test set (except for gpt4o model, where on test set, setting $\alpha = 0.4$ is enough to achieve a comparable accuracy to Adaptive-RAG). Thus, users can rely on validation-based tuning of $\alpha$ for effective deployment.

### C.3 MORE COMPARISONS TO POTENTIAL BASELINES

As our work focus on user controllability, while none of the related works (such as Wang et al. (2023); Tang et al. (2024)) provides this property, we compare them in the appendix for completeness. Moreover, we also compare with a additional baseline, where instead of training a classifier, we directly prompt the LLM to decide the complexity of the retrieval, which we coin as "direct

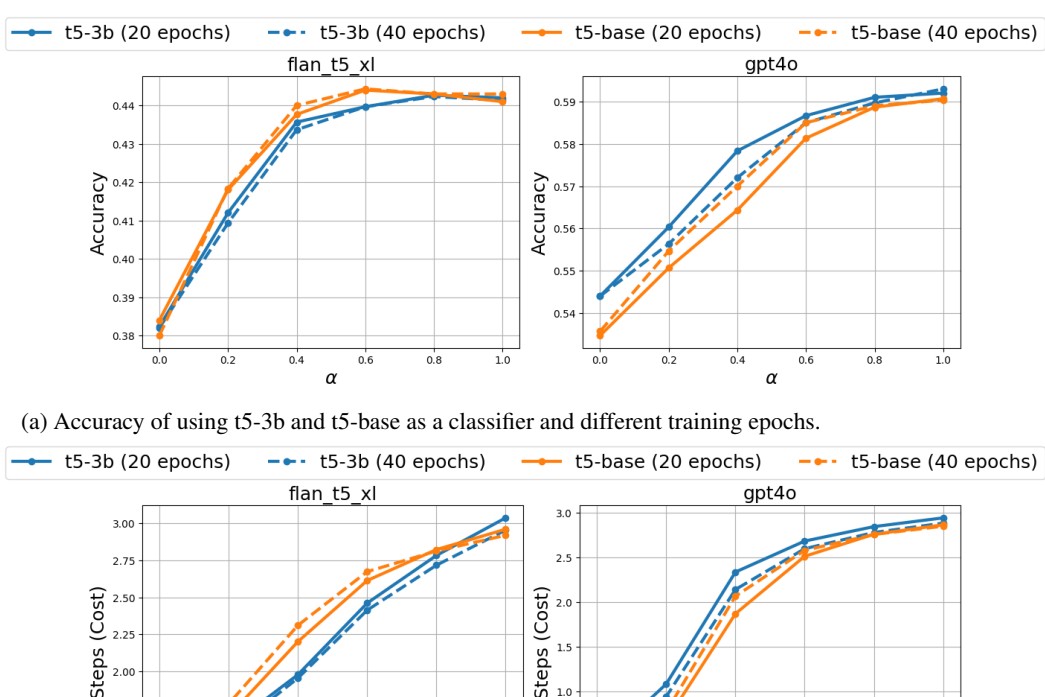

(a) Accuracy of using t5-3b and t5-base as a classifier and different training epochs.

(b) Total retrieval steps (cost) of using t5-3b and t5-base as a classifier and different training epochs.

Figure 8: Accuracy and total steps of t5-3b and t5-base as a classifier and different training epochs.

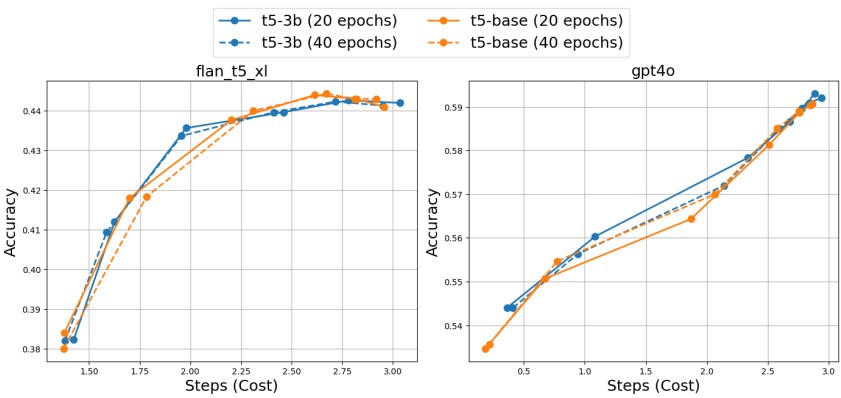

Figure 9: Accuracy-Cost trade-off plot of using t5-3b and t5-base as a classifier with different training epochs.

prompting" baseline. In Table 5, we report the performance of MBA-RAG Tang et al. (2024) and self-RAG Wang et al. (2023). However, note that none of these baselines provide user controllability in their trade-off of accuracy and speed. We notice that direct prompting achieves good performance. However, as the main contributions of this paper is user controllability, which the direct prompting approach doesn't provide, we didn't compare it in the main paper. Nevertheless, we think it would be a good approach to explore more and a potential replacement for adaptive retrieval. (Still, it

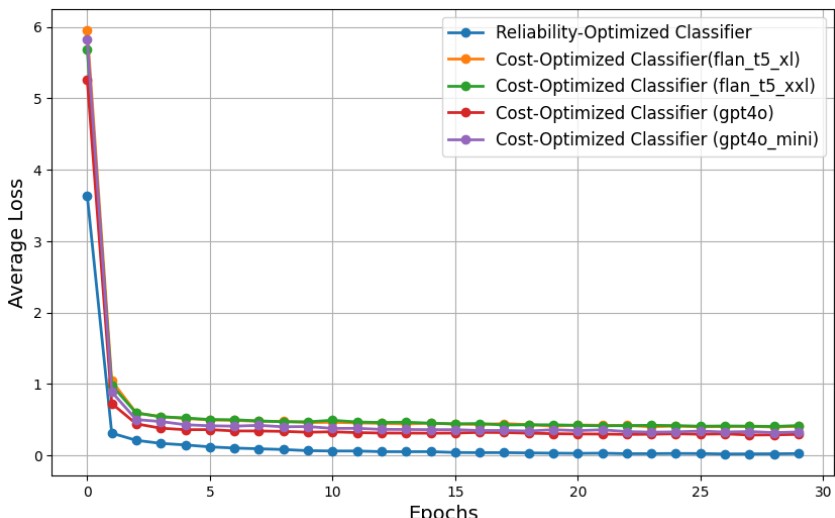

Figure 10: Average training loss v.s. training epochs for cost optimized classifier for different models and reliability optimized classifier.

| Models | flan-t5-xl | | flan-t5-xxl | | gpt4o-mini | | gpt4o | |
|---|---|---|---|---|---|---|---|---|
| | Acc | Steps | Acc | Steps | Acc | Steps | Acc | Steps |
| **Static Retrieval** | | | | | | | | |
| No Retrieval | 0.160 | 0.0 | 0.193 | 0.0 | 0.408 | 0.0 | 0.527 | 0.0 |
| Single-step Retrieval | 0.389 | 1.0 | 0.419 | 1.0 | 0.477 | 1.0 | 0.539 | 1.0 |
| Multi-step Retrieval | 0.437 | 4.7 | 0.452 | 2.1 | 0.533 | 5.6 | 0.595 | 4.2 |
| **Adaptive-RAG** | | | | | | | | |
| AdaptiveRAG | 0.427 | 2.3 | 0.439 | 1.3 | 0.510 | 2.6 | 0.585 | 2.0 |
| **Flare-Aug(Ours)** | | | | | | | | |
| $\alpha = 0.0$ | 0.388 | 1.3 | 0.385 | 0.9 | 0.445 | 0.5 | 0.533 | 0.2 |
| $\alpha = 0.2$ | 0.421 | 1.6 | **0.444** | **1.2** | 0.490 | 1.7 | 0.557 | 0.7 |
| $\alpha = 0.4$ | **0.439** | **2.2** | 0.461 | 1.4 | 0.524 | 2.8 | 0.575 | 2.2 |
| $\alpha = 0.6$ | 0.441 | 2.6 | 0.465 | 1.5 | 0.532 | 3.2 | 0.584 | 2.6 |
| $\alpha = 0.8$ | 0.439 | 2.8 | 0.463 | 1.6 | 0.533 | 3.4 | 0.589 | 2.8 |
| $\alpha = 1.0$ | 0.441 | 3.0 | 0.463 | 1.6 | 0.534 | 3.5 | 0.592 | 2.9 |

Table 3: Accuracy and average steps of different retrieval strategies on validation dataset.

doesn't provide user controllability). It might be an important future direction to use prompting to decide retrieval strategy while adding user controllability.

| Models | flan-t5-xl | | flan-t5-xxl | | gpt4o-mini | | gpt4o | |
|---|---|---|---|---|---|---|---|---|
| | Acc | Steps | Acc | Steps | Acc | Steps | Acc | Steps |
| **Static Retrieval** | | | | | | | | |
| No Retrieval | 0.154 | 0.0 | 0.184 | 0.0 | 0.402 | 0.0 | 0.512 | 0.0 |
| Single-step Retrieval | 0.386 | 1.0 | 0.418 | 1.0 | 0.491 | 1.0 | 0.553 | 1.0 |
| Multi-step Retrieval | 0.423 | 4.6 | 0.443 | 2.1 | 0.543 | 5.6 | 0.599 | 4.1 |
| **Adaptive-RAG** | | | | | | | | |
| AdaptiveRAG | 0.414 | 2.3 | 0.437 | 1.3 | 0.528 | 2.6 | 0.591 | 2.0 |
| **Flare-Aug (Ours)** | | | | | | | | |
| $\alpha = 0.0$ | 0.381 | 1.3 | 0.388 | 0.8 | 0.445 | 0.5 | 0.520 | 0.2 |
| $\alpha = 0.2$ | 0.408 | 1.7 | 0.443 | 1.2 | 0.498 | 1.6 | 0.549 | 0.6 |
| $\alpha = 0.4$ | 0.425 | 2.2 | 0.455 | 1.4 | 0.541 | 2.8 | 0.593 | 2.2 |
| $\alpha = 0.6$ | 0.427 | 2.6 | 0.452 | 1.5 | 0.547 | 3.2 | 0.598 | 2.5 |
| $\alpha = 0.8$ | 0.428 | 2.8 | 0.451 | 1.6 | 0.550 | 3.4 | 0.600 | 2.7 |
| $\alpha = 1.0$ | 0.430 | 2.9 | 0.450 | 1.6 | 0.550 | 3.5 | 0.601 | 2.8 |

Table 4: Accuracy and average steps of different retrieval strategies on test dataset.

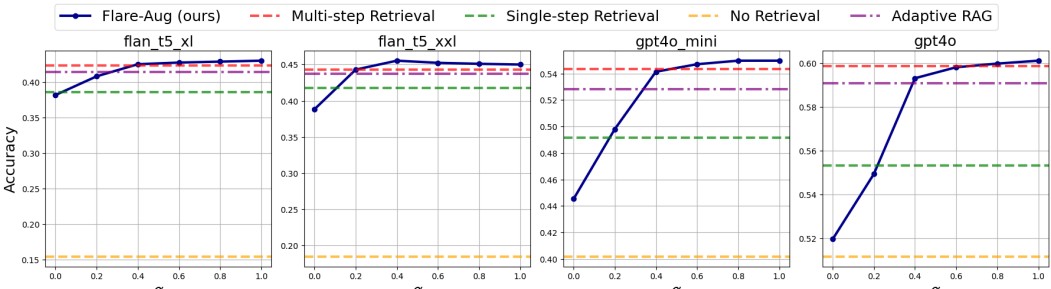

Figure 11: Retrieval accuracy of different approaches on test set.

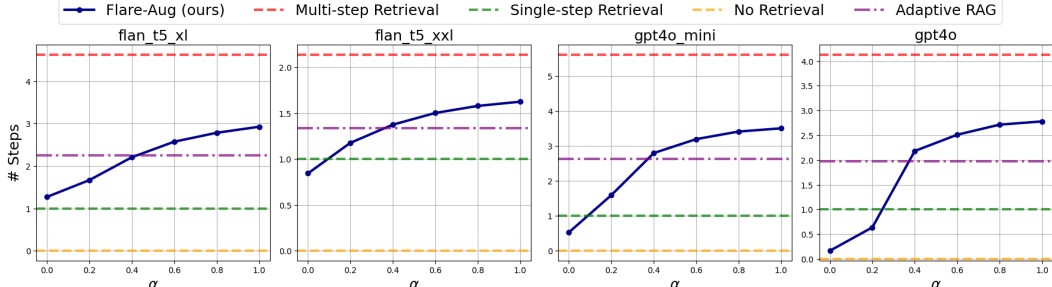

Figure 12: Total number of steps (cost) of different approaches on test set.

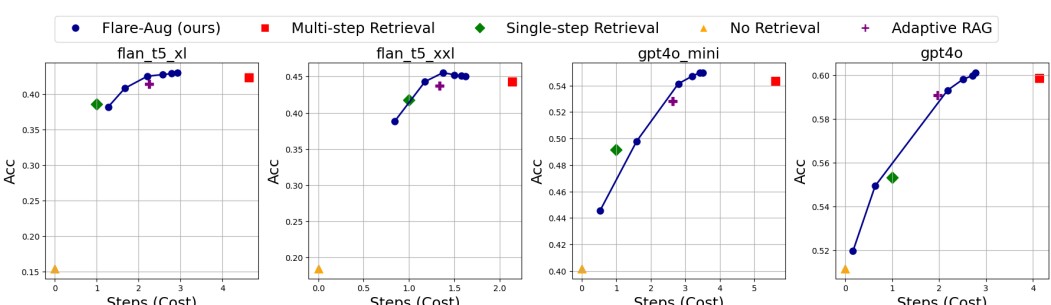

Figure 13: Accuracy-Cost plot of different approaches on test set.

| Models | Acc | Steps |
|---|---|---|
| **Static Retrieval** | | |
| No Retrieval | 0.160 | 0.0 |
| Single-step Retrieval | 0.389 | 1.0 |
| Multi-step Retrieval | 0.437 | 4.7 |
| **Adaptive-RAG** | | |
| AdaptiveRAG | 0.427 | 2.3 |
| **Added Baselines** | | |
| MBA-RAG (DistilBert) | 0.435 | 1.80 |
| MBA-RAG (T5-Large) | 0.434 | 1.89 |
| Self-RAG (LLaMA2) | 0.316 | 0.72 |
| Direct prompting | 0.381 | 0.19 |
| **Flare-Aug (Ours)** | | |
| $\alpha = 0.0$ | 0.388 | 1.3 |
| $\alpha = 0.2$ | 0.421 | 1.6 |
| $\alpha = 0.4$ | 0.439 | 2.2 |
| $\alpha = 0.6$ | 0.441 | 2.6 |
| $\alpha = 0.8$ | 0.439 | 2.8 |
| $\alpha = 1.0$ | 0.441 | 3.0 |

Table 5: Results for additional Baselines. The results for MBA-RAG and Self-RAG are from the original paper, using DistillBert, T5-Large, LLaMA2 as a backbone, respectively, while the rest of the results uses t5-large as a LLM backbone.

