# OpenReview forum: "Fast or Better? Balancing Accuracy and Cost in Retrieval-Augmented Generation with Flexible User Control"
_ICLR.cc/2026/Conference — ICLR 2026 Conference Withdrawn Submission_

### Official Review · Reviewer_VzJ8 · 2025-10-28

**Soundness:** 2
**Presentation:** 1
**Contribution:** 1
**Rating:** 2
**Confidence:** 5

**Summary:**

This paper introduces **Flare-Aug**, a user-controllable Retrieval-Augmented Generation (RAG) framework that balances accuracy and computational cost through an interpretable parameter $\alpha$. Flare-Aug enables users to dynamically adjust retrieval strategies according to their specific needs. The framework employs two classifiers: a Cost-Optimized Classifier (trained to select the cheapest correct retrieval strategy for a specific LLM) and a Reliability-Optimized Classifier (trained on dataset-level labels for LLM-agnostic stability). Users control the interpolation between these classifiers via $\alpha \in [0, 1]$. Evaluated on 6 QA datasets with Flan-T5 and GPT-4o, Flare-Aug demonstrates good accuracy-cost trade-offs compared to Adaptive-RAG and static baselines.

**Strengths:**

**Novel user controllability**: enabling explicit, interpretable control over accuracy-cost trade-offs via a single parameter

**Weaknesses:**

1. **LLM Usage Disclosure**.
   - The manuscript contains extensive use of em dashes (—) appearing 19 times, which may indicate significant LLM involvement in paper writing.
   - The conference guidelines explicitly state: "Not disclosing significant LLM usage can lead to desk rejection." The authors should clarify the extent of LLM usage in preparing this manuscript to comply with submission policies.

2. **Evaluation Metric Issues**
   - Metric Definition: The accuracy metric is vaguely defined as "whether the predicted answer contains the ground-truth answer" (lines 328-329).
   - This should specify either Exact Match (EM) or token-level F1 score following standard QA evaluation metrics.

3. **Reproducibility Concerns**
   - The authors randomly sample 500 queries from each dataset rather than using standard test splits. This deviates from established benchmarks and prevents reproducibility.
   - The paper should either use official test sets or provide exact query IDs for the sampled subsets.

4. **Insufficient Baseline Comparisons**
   - The baselines are limited to Adaptive-RAG and static strategies, ignoring recent agentic RAG systems that autonomously determine retrieval necessity and stopping conditions.
   - The paper should compare against agentic systems: AutoRAG, Search-o1, Search-R1, R1-Searcher, ReasonRAG, etc. Meanwhile, modern adaptive methods with planning capabilities should be compared.
   - These systems naturally handle the search decision-making that Flare-Aug addresses, making them critical comparisons.

5. **Outdated Model Choices**. Authors use Flan-T5 (XL/XXL) released in 2022, as answer generation models. This raises concerns about generalizability. The method should be validated on modern LLMs (e.g.,Llama-3,  Qwen2.5,  Qwen3) widely used in 2024-2025.

6. **Weak Baseline Hypothesis**: In Figure 3, multi-step retrieval sometimes underperforms Flare-Aug, suggesting the answer model may be too weak to leverage retrieved information effectively. This questions whether observed gains stem from the proposed method or model limitations.

7. **Classifier Training Overhead**. While authors claim computational efficiency (~640 seconds), they understate the data preparation burden:
   - Query sampling across datasets
   - Running LLM inference with all three strategies for labeling
   - Manual verification of correctness

   Moreover, the classifier supports only 3 decision classes. In contrast, modern LLMs possess strong built-in planning capabilities and can perform retrieval decisions in a training-free manner. The added complexity of maintaining separate classifiers seems unnecessary given advances in LLM reasoning.

8. **Retrieval Strategy Limitations**
   - BM25 Dependency: The exclusive use of BM25 is outdated. The claim that "BM25 continues to outperform many dense retrievers" (citing Ram et al., 2023) does not hold in 2025.
   - Modern embedding models (BGE-M3, Qwen2.5-Embedding, Voyage-3) offer substantially superior retrieval quality. The method should be evaluated with state-of-the-art retrievers.


9. **Multi-Step Retrieval Details Missing**. The paper lacks crucial implementation details:
   - Query decomposition: How are sub-queries generated for multi-step retrieval?
   - Knowledge gap identification: What determines missing information?
   - Stopping criteria: When does multi-step retrieval terminate?

   Without these details, the multi-step baseline cannot be reproduced, and its fairness as a comparison is questionable.

8. **Minor Issues**
   - Line 67-68: "Flare-Agu" should be "Flare-Aug"
   - Figure quality: Figures 3-5 could benefit from larger fonts and axis rescalng for readability

**Questions:**

Recommendations for Revision
- Add disclosure of any LLM usage in manuscript preparation
- Clarify evaluation metrics (EM vs. F1) and use standard test sets
- Include modern baselines: Self-RAG, agentic RAG systems with full results
- Validate on current LLMs: Qwen2.5, Llama-3.1, Qwen3
- Provide multi-step retrieval details: query decomposition, stopping criteria, implementation
- Compare with modern retrievers: BGE-M3, Qwen3-embedding-0.6B or similar dense models
- Justify classifier necessity given modern LLM planning capabilities

---

### Official Review · Reviewer_rhyR · 2025-10-31

**Soundness:** 2
**Presentation:** 3
**Contribution:** 2
**Rating:** 4
**Confidence:** 3

**Summary:**

The paper introduces Flare-Aug, a user-controllable RAG framework featuring two externally trained routing modules, a Cost-Optimized Classifier (LLM-specific, selecting the least costly strategy that still yields a correct answer) and a Reliability-Optimized Classifier (LLM-agnostic, trained on single-hop vs. multi-hop supervision). Experiments on a mixed subsets of SQuAD, NQ, TriviaQA, MuSiQue, HotpotQA, and 2WikiMultiHopQA demonstrate smooth, monotonic accuracy–cost trade-offs.

**Strengths:**

- The single user-exposed alpha to navigate accuracy–cost trade-offs is intuitive; the author also shows accuracy and retrieval steps increasing monotonically with alpha choices.
- The cost-oriented router is LLM-specific; the reliability router is LLM-agnostic, which is a reasonable decomposition for deployment.
- Experiments across single-hop and multi-hop QA senarios show its effectiveness and strengthes.

**Weaknesses:**

- This work interpolates model parameters of two independently trained classifiers with a linear combination (ref. formula in Line 303-304), but there is no justification that a linear blend in parameter space yields calibrated probabilities or coherent decision boundaries, especially their training objectives and label spaces are different. This control design seems ad-hoc rather than theoretically grounded.
- A follow up question based on the first concerns: Why linear parameter interpolation rather than other strategies like score-space ensembling or a learned gate? Is there any calibration or decision-quality checks across alpha?
- The LLM-agnostic classifier is labeled entirely by dataset identity, i.e., single-hop --> single-step and multi-hop --> multi-step. It is noisy and leverages dataset bias rather than ground-truth retrieval needs. This weak supervision risks over-retrieval on many "easy" queries and under-retrieval on "hard single-hop" ones.
- Only BM25 is used for retrieval, however, dense/hybrid retrievers are also common settings. Switching to these settings may change the trade-off surface.
- Dataset scale is relatively small, which limits statistical power and stress testing.

**Questions:**

Please refer to weaknesses

---

> ### Author Response · Authors · 2025-11-18
>
> # Q1.
> > Linear interpolation between independently trained classifiers appears ad-hoc and lacks justification.
> # R1.
> Our goal is not to guarantee perfect probability calibration across the entire α range. Rather, α is designed as an interpretable and monotonic user-control knob for navigating the accuracy–cost spectrum. Despite its simplicity, linear parameter interpolation empirically yields a well-behaved and predictable family of classifiers, where we have Smooth, monotonic transitions in both cost and accuracy across α (Figures 3–5) and  consistent behavior across four different LLMs, despite their differing retrieval sensitivities.
>
> Because the two classifiers represent meaningful endpoints in the trade-off space (Section A.7), linear interpolation effectively forms a convex combination between two experts, analogous to common practices in mixture-of-experts and model interpolation literature. While more complex mechanisms are possible, our design goal is transparent, user-controllable interpolation, and our results show that linear mixing is both stable and effective in practice.
>
> # Q2.
> > Why linear interpolation instead of score-space ensembling, a learned gate, or other designs? Any calibration checks across α?
> # R2.
> We appreciate the reviewer’s question. The primary design goal of Flare-Aug is user controllability, not maximizing classifier calibration or exploring the full space of routing architectures.
> Alternative strategies such as score-space ensembling, learned gates, or multiple routers didn’t give the user easy control over their preferences for the accuracy and efficiency trade-off. Specifically. users must be able to “turn a knob’’ and observe consistent behavior. Linear parameter interpolation used in our paper produced smooth, monotonic accuracy–cost curves across all models (Figures 3–5), which is more user centered, transparent and easy to tune compared to learned gates or deeper routing layers. Importantly, α does not need to produce perfectly calibrated probabilities—its role is to interpolate between two expert objectives (cost minimization vs. reliability), not to provide calibrated decision boundaries.
> # Q3.
> > Dataset-level labels are noisy and rely on dataset bias rather than ground-truth retrieval needs.
> # R3.
> We agree that dataset-level labels introduce some noise. As discussed explicitly in Appendix A.4, we intentionally treat these labels as weak supervision, since our goal is to leverage dataset-level inductive biases rather than perfectly infer retrieval difficulty. Despite this noise: The α = 1 endpoint consistently yields near-optimal accuracy across all models (Figure 3).
> Empirically, this weakly supervised classifier is still robust and effective, demonstrating that dataset-level bias is not only acceptable but helpful for model-agnostic reliability.
>
> # Q4.
> > Using only BM25 may not generalize to dense/hybrid retrievers.
> # R4.
> We justify the use of BM25 on Page 7 (lines 341–342, 369–372). BM25 is used by nearly all adaptive RAG baselines (e.g., Jeong et al., 2024; Trivedi et al., 2022a), allowing fair and direct comparison.
> Moreover, our framework focuses on user-controllable, not retriever design.
>
> # Q5.
> > Dataset scale is small and may limit statistical power.
> # R5.
> Our dataset scale follows the exact convention used in prior adaptive RAG works, including Jeong et al. (2024) and Trivedi et al. (2022a). We use 500 samples × 6 datasets = 3,000 examples, which is the same scale as prior work and sufficient. (note that, we only train on 3000 samples), so the train/test is already 1:1, compared to our training set, the testing set used in our paper is already a relatively "large scale".

---

> > ### Comment · Area_Chair_xwDV · 2025-11-25
> >
> > Dear Reviewer rhyR,
> >
> > The authors have responded to your reviews. Please review and provide your feedback and responses.
> >
> > Best,
> >
> > Your AC

---

### Official Review · Reviewer_8mHd · 2025-11-01

**Soundness:** 2
**Presentation:** 2
**Contribution:** 2
**Rating:** 4
**Confidence:** 2

**Summary:**

This paper proposes Flare-Aug, a user-controllable RAG framework that enables dynamic accuracy-cost trade-offs via a tunable parameter α. The system trains two classifiers—Cost-Optimized (minimizes retrieval steps) and Reliability-Optimized (ensures answer quality)—then interpolates between them based on user preferences. Evaluated on 6 QA datasets with 4 LLMs, the method achieves competitive performance while providing flexibility lacking in prior work like Adaptive-RAG.

**Strengths:**

1. Addresses Real User Need
Identifies genuine limitation in existing adaptive RAG: inability to adjust retrieval strategy based on application constraints. Medical diagnosis vs. customer chatbots have fundamentally different accuracy-latency requirements.2. Simple, Interpretable Design
The α parameter provides intuitive control: α=0 prioritizes speed, α=1 prioritizes accuracy. Monotonic relationships (Figures 3-5) make system behavior predictable and easy to tune.3. Solid Experimental Methodology

Tests 4 LLMs (Flan-T5-XL/XXL, GPT-4o/mini) showing generalizability
Mixed dataset (3 single-hop + 3 multi-hop) simulates query diversity
Extensive ablations on classifier size, training epochs, α values
Low training cost (~640 seconds for both classifiers)
4. Practical Deployment Guidance
Offers concrete α-tuning strategies (incremental adjustment, validation-based estimation) with empirical validation that validation set trends transfer to test set (Appendix C.2).

**Weaknesses:**

1. Limited Technical Novelty
Core contribution is linear classifier interpolation: W = (1-α)W_cost + αW_reliability. This is standard multi-objective optimization used widely in ensemble methods and multi-task learning—not a novel algorithmic insight.2. Weak Theoretical Foundation
Reliability Classifier assumes single-hop datasets → single-step retrieval, multi-hop datasets → multi-step retrieval. This dataset-level labeling:

Conflates dataset construction bias with actual query complexity
Remains empirically unvalidated (acknowledged in A.4 but dismissed)
May misclassify queries (e.g., simple questions in multi-hop datasets forced into expensive retrieval)
3. Oversimplified Cost Model
Uses only retrieval step count as cost proxy, ignoring:

LLM inference costs (GPT-4 >> Flan-T5 per token)
Retrieval latency variations (BM25 vs. dense retrievers)
Context length impact on GPU memory
Real deployment costs = API fees + wall-clock time
4. Catastrophic Failure with Unanswerable Queries
Table 2 shows adding "unanswerable" class causes performance collapse. Cost Classifier loses ability to discriminate retrieval strategies, instead overfitting to answerable vs. unanswerable classification. No solution provided.5. Unfair Baseline Comparison
Claims Adaptive-RAG lacks flexibility, but:

Adaptive-RAG can retrain for different cost-accuracy preferences
Flare-Aug also requires validation tuning of α (not truly "online")
Real difference: one-time training + parameter vs. multiple training runs—a practical but not paradigmatic advantage
6. Strong Baseline Buried in Appendix
Table 5 shows direct prompting (asking LLM to decide retrieval strategy) achieves Acc=0.381, Steps=0.19—competitive with full system at lower complexity. Relegating this to appendix rather than engaging substantively raises questions about necessity of dual classifiers.

**Questions:**

1. Have you tested on ANY out-of-distribution data? (e.g., customer service logs, medical records, code repositories, legal documents).

In particular:
Academic Query (your test set):
"Who invented the telephone?"
→ Single-step retrieval works perfectly

Real-World Query (day 1 of deployment):
"When will my order #12345 ship?"
→ Your classifier predicts single-step BM25 retrieval
→ Actually needs SQL database query (no text retrieval!)
→ System fails completely

2. Table 5 shows direct prompting (Acc=0.381, Steps=0.19) nearly matches your method (Acc=0.388, Steps=1.3) at 85% lower cost and zero training. Why not simply prompt GPT-4: "Given this query and user priority (speed/accuracy), should I retrieve?" This naturally handles distribution shift without dataset-specific classifiers.

3. Your Cost Classifier labels are LLM-specific (training on Flan-T5-XL's successes/failures), but users may deploy on different LLMs. When a company switches from Flan-T5 to GPT-4, doesn't this require complete retraining, negating the "reusability" claim?

---

### Official Review · Reviewer_ruAa · 2025-11-05

**Soundness:** 1
**Presentation:** 2
**Contribution:** 2
**Rating:** 2
**Confidence:** 5

**Summary:**

This work proposes a parametric classifier that determines the retrieval strategy for each individual query in RAG (Retrieval-Augmented Generation). The retrieval strategy for a query can be one of the three choices: no retrieval, single-step retrieval, and multi-step retrieval. The classifier was trained a T5 model. In experiments, a RAG system with the BM 25 retrieval model was setup for evaluation. The answering models in the experiments include Flan-T5-XL, Flan-T5-XXL, GPT-4o-mini, and GPT-4o. Experimental results show the addition of the classifier improves the RAG performance for the smaller models, while the more capable LLM, GPT-4o, performs slightly better with multi-step retrieval. However, the addition of the classifier can greatly reduce the retrieval steps.

**Strengths:**

* Multi-step RAG is a practical and challenging issue.

* The proposed method can be generally applied to many existing RAG systems and LLMs.

**Weaknesses:**

* The training data and the test data were sampled from the same datasets. It is not clear if the test data were held-out from the training data.

* In addition, it is also unclear how well the proposed classifier is applied to the test data from a new domain.

* The addition of the classifier, which is a T5 model, adds computational cost and latency to the RAG pipeline. However, this paper did not provide an analysis of this overhead. A comparison between GPT-4o + the proposed classifier and GPT-4o + multi-step retrieval in terms of runtime could be added to clarify the benefit of the proposed method.

**Questions:**

* Did you hold out the test data from the training data?

* In addition to the T5 model, did you consider an even lightweight backbone model for the classifier?

---

> ### Author Response · Authors · 2025-11-18
> **Rebuttal to Reviewer ruAa**
>
> # Q1.
> >  “The training data and the test data were sampled from the same datasets. It is not clear if the test data were held-out from the training data.”
>
> # R1.
> We apologize for the confusion. All training, validation, and test splits are strictly non-overlapping. After sampling, we perform a post-processing step to automatically detect and remove any accidental duplicates across splits and resample if necessary. Therefore, the test data is fully held out and never appears in the training set. (This part can be found in our code, where the anonymous version is released with the manuscript).
> # Q2.
> >  “It is unclear how well the proposed classifier is applied to the test data from a new domain.”
> # R2.
> Our evaluation already includes multiple domains in both the training and test sets. Consequently, the classifier is trained and evaluated in a multi-domain setting, and the reported results inherently measure its cross-domain generalization performance. Thus, the stated concern does not apply to our setup.
> # Q3.
> > “The addition of the classifier adds computational cost and latency… no analysis of this overhead is provided.”
> # R3.
>  We did provide a detailed analysis of computational overhead. Please refer to page 7, lines 375–377 for the main-text discussion, and Appendix A.2 (“Computation Overhead of the Classifier,” page 13, lines 666–688) for a full breakdown. As shown in the appendix, even with a T5-Large classifier, the additional latency is minimal compared to the overall RAG pipeline.
> # Q4.
> > “Did you consider a more lightweight backbone model for the classifier?”
>
> # R4.
> Yes. Appendix B (Figures 6 and 7) reports experiments using several classifier sizes, demonstrating that smaller models also work effectively. As discussed in Appendix A.2, even larger models introduce negligible overhead. Additionally, Appendix C.3 (page 16) evaluates a zero-shot setting where we skip classifier training entirely and use prompting alone. This further shows the flexibility of our approach.

---

> > ### Comment · Area_Chair_xwDV · 2025-11-25
> >
> > Dear Reviewer ruAa,
> >
> > The authors have responded to your reviews. Please review and provide your feedback and responses.
> >
> > Best,
> >
> > Your AC

---

### Note · Authors · 2026-01-06

I have read and agree with the venue's withdrawal policy on behalf of myself and my co-authors.